# Epi-attention : Adaptive Context-Aware Attention for Dynamic Feature Relevance in Neural Networks

## Abstract

In this paper, we introduce Epi-Attention, a novel context-aware attention mechanism designed to enhance the relevance of features in neural networks by incorporating external contextual information. Unlike traditional attention mechanisms that rely solely on the input sequence, Epi-Attention dynamically adjusts the significance of features based on additional evidence provided by external contexts. This approach allows the model to emphasize or diminish the relevance of specific features, leading to better capture and reflect the internal properties of specific classes. This mechanism provides a nuanced interpretation of feature relevance that aligns with domain knowledge, enabling the model to focus on contextually significant features in a way that resonates with expert understanding. We formalize the problem and present two variants of the proposed mechanism: Scaled Dot-Product Epi-Attention and Self-Epi-Attention, both of which re-evaluate feature importance considering either external or internal information, respectively. By leveraging the dynamic aspect of Epi-Attention, models can highlight local correlations that are characteristic of certain classes, offering a more transparent and interpretable decision-making process compared to global correlations favorized by classical approaches such as Decision trees, Logistic regression and Neural Networks. We demonstrate the efficency of Epi-Attention through three different applications (dynamic feature relevance, processing mixed datatypes and multi-source datasets) with respectively benchmark datasets, including the Wisconsin Breast Cancer, Bank Marketing and ABIDE-II datasets. Our results show significant improvements in model interpretability over traditional models that aligns with domain knowledge. Furthermore, we discuss the potential of Epi-Attention for enhancing explainability in complex machine learning tasks, paving the way for more robust and transparent neural network architectures.

## 1 Introduction

Attention mechanisms Vaswani et al. (2017) have become a core component in deep learning, significantly enhancing the ability of models to focus on the most relevant aspects of input data Hassanin et al. (2024); Xiao et al. (2024); Bo et al. (2024). However, conventional attention mechanisms, including those applied in models like transformers Vaswani et al. (2017), often assign static importance weights across the entire dataset. This global approach can be misaligned with real-world scenarios involving heterogeneous data, where local correlations, class-specific properties, and external context are critical. As a result, traditional attention methods may fail to capture the nuanced patterns needed for tasks involving mixed data types, class imbalance, or varying subpopulations.

In Addition, explainability is crucial in deploying machine learning models in domains such as healthcare and finance, where decision-making must be transparent and aligned with expert knowledge. Traditional models, including logistic regression, MLPs, decision trees, and random forests, often suffer from several challenges:

1. **Static Feature Importance:** These models apply static importance weights to features across all data points, failing to capture class-specific insights.

2. **Missed Local Correlations:** Domain experts typically focus on local correlations—feature relationships within specific classes or subpopulations—that reflect the internal properties of each class. Traditional models struggle to capture these finer details, missing valuable insights that could improve interpretability.

3. **Bias in Imbalanced Datasets:** Importance weights often align with global correlations, reflecting the properties of the most represented class. This global focus can obscure critical features in underrepresented classes, leading to biased models and misalignment with domain expertise.

To address these limitations, we introduce Epi-Attention, a dynamic, context-aware attention mechanism that adjusts feature relevance based on both internal data structure and external contextual information. Epi-Attention allows models to dynamically emphasize or de-emphasize specific features in response to domain-specific contextual cues. Additionally, we propose Self-Epi-Attention, which focuses on internal consistency within a given input, further refining feature relevance without relying on external context.

Our main contributions are as follows:

- We propose Epi-Attention, a mechanism that leverages internal and external contexts to dynamically adjust feature relevance, enhancing model adaptability and interoperability.
- We introduce Self-Epi-Attention, which focuses on capturing class-specific internal correlations, further improving the interpretability of learned features.
- We demonstrate the applicability of Epi-Attention across a variety of domains, including medical diagnosis, marketing, and neuroimaging, where challenges such as class imbalance, mixed data types, and site-specific variability are prevalent.

We validate the proposed mechanisms through experiments on several tasks and benchmark datasets. Our results show that Epi-Attention significantly improves model transparency and decision-making, particularly in scenarios requiring context-specific or class-specific feature weighting.

## 2 RELATED WORK

Attention mechanisms have become a crucial component in various deep learning models for tasks such as machine comprehension, action recognition, emotion recognition, and natural language processing. Traditionally, attention mechanisms have been used to focus on specific parts of the input data and summarize it with fixed-size vectors Britz et al. (2017). However, recent advancements have introduced more sophisticated attention mechanisms that take into account contextual information to improve performance. For instance, in Seo et al. (2017), a Bi-Directional Attention Flow (BIDAF) network has been introduced. It utilizes a multi-stage hierarchical process to represent context at different levels of granularity and incorporates a bi-directional attention flow mechanism to obtain a query-aware context representation without premature summarization. Similarly, Liu et al. (2017) and Liu et al. (2018) proposed Global Context-Aware Attention LSTM networks for 3D action recognition and skeleton-based action recognition, respectively. These networks selectively focus on informative joints in action sequences with the assistance of global contextual information, improving attention representation iteratively Pang et al. (2023); Liu et al. (2021). In the realm of natural language processing, Xing et al. (2017) introduced a Topic Aware Sequence-to-Sequence (TA-Seq2Seq) model that utilizes joint attention mechanisms to summarize hidden vectors of input messages and synthesize topic vectors from topic words obtained from a pre-trained LDA model Zhao et al. (2011). Furthermore, Yu et al. (2022) highlighted the limitations of traditional Bi-Attention mechanisms in explicitly modeling interactions between contexts, queries, and keys of target sequences. To address this, they proposed a Tri-Attention framework that incorporates context as the third dimension in calculating relevance scores, enhancing attention performance in natural language processing tasks Vaswani et al. (2017). Moreover, attention mechanisms have also been applied in emotion recognition tasks. Lee et al. (2019) introduced Context-Aware Emotion Recognition Networks (CAER-Net) that leverage both human facial expressions and contextual information in a joint manner for improved emotion recognition. Additionally, Zhong et al. (2019) proposed a Knowledge-Enriched Transformer (KET) for emotion detection in textual conversations, where contextual utterances are interpreted using hierarchical self-attention and external commonsense

knowledge is dynamically leveraged using a context-aware affective graph attention mechanism. In summary, attention mechanisms have evolved from simple uni-directional approaches to more sophisticated context-aware mechanisms that consider global context, topics, and external knowledge to enhance performance across various domains such as machine comprehension, action recognition, emotion recognition, and natural language processing. The introduction of Tri-Attention frameworks represents a significant advancement in explicitly modeling interactions between contexts, queries, and keys, leading to improved attention performance in complex tasks.

## 3 NOTATIONS AND DEFINITIONS

This section introduces the notations used throughout this paper.

The set $\mathcal{D}$ represents the set of observations. Each element $x \in \mathcal{D}$ corresponds to a specific recording of an individual from the population under study. Also, we consider the set $\mathcal{M}$ that represents the metadata available, called context, for each element $x \in \mathcal{D}$ and the function $\chi$ that maps elements from $\mathcal{D}$ to elements from $\mathcal{M}$.

$$\chi \colon \mathcal{D} \to \mathcal{M} \\ x \mapsto c \tag{1}$$

Both observations and metadata are assumed to be in raw sequential format i.e. $x = [x^1, x^2, \cdots, x^j, \cdots, x^p]$ (resp. $c = [c^1, c^2, \cdots, c^l]$) with $p$ (resp. $l$) the length of the sequence $x$ (resp. $c$). and $x^j, j = 1, ..p$ could be of different data types.

We assume that we have observations corresponding to $N$ individuals, represented by the subset $X = \{x_i; i = 1, \ldots, N\} \subset \mathcal{D}$ and $\mathcal{C} = \{c \in \mathcal{M}; c = \chi(x)\} \subset \mathcal{M}$. Of course, it is assumed that $X$ is representative of the population under study and will serve as the training set for estimating the parameters of various models.

Processing the raw sequence $x \in X$ can pose challenges, particularly when dealing with mixed data types (images, words,...). Several methods aim, in the first instance, to encode/embed the information contained in $x_i$ within a vector space to overcome these challenges de Kok et al. (2024); Sahoo & Chakraborty (2020); they do so by :

- defining a sequence to one (seq2one) embedding function $f$ that encodes the hole sequence $x \in X$ as one vector $v \in \mathbb{R}^d$ :
$$f \colon X \to \mathbb{R}^d \\ x \mapsto v \tag{2}$$

- or defining a sequence to sequence (seq2seq) embedding function $g$ that encodes each element $x^j \in x$ (with $x \in X$) as a vector $v^j \in \mathbb{R}^d$, so :
$$g \colon X \to \mathbb{R}^{d \times p} \\ x \mapsto [v^j]_{j=1}^p \tag{3}$$

## 4 PROBLEM FORMULATION

Let $X = \{x_i; i = 1, \ldots, N\} \subset \mathcal{D}$ the dataset under study and $\mathcal{C} = \{c \in \mathcal{M}; c = \chi(x)\} \subset \mathcal{M})$ the related context set.

In Machine Learning, it is common to process the data observation $x_i \in X$ i.e. the raw sequence $x_i = [x_i^1, x_i^2, \cdots, x_i^j, \cdots, x_i^p]$ by assuming that all elements $x_i^j$ of $x_i$ are independent and relevant as inputs.

There are situations where additional information (context), $c_i = \chi(x_i)$, becomes available. This external information raises questions about the relevance of certain elements of $x_i$ based on the evidence at hand. i.e. knowing some external information $c_i$ can enhance (or reduce) the relevance of some variables for a specific observation $x_i$. This essentially requires us to assess the plausibility

of the hypothesis $H_i^j$ related to the $j^{th}$ element of observation $x_i$ based on the given evidence $E_i$ with :

- $H_i^j : x_i^j \; is \; relevant \; for \; x_i$ (hypothesis)

- $E_i : \; c_i \; is \; additional \; information \; about \; x_i$ (evidence)

$\forall i = 1, \ldots, N$ and $j = 1, \ldots, p$

Consider an example where we aim to predict the outcome of a bank marketing campaign using several numerical features such as 'age', 'balance', 'last contact duration', ... noted respectively $[x^1, x^2, x^3, \ldots]$.

Experts[1] state that the feature $x^3$ ('last contact duration') "highly affects the output target...", suggesting its high relevance.

However, the relevance of $x^3$ can be influenced by an additional piece of external information $c^1$ which asks whether the client was contacted before. If $c^1 = 1$ ("yes, he was contacted"), the relevance of $x^3$ remains high. Conversely if $c^1 = 0$ ("no, he wasn't contacted before"), the relevance of $x^3$ decreases significantly. In this case, a well-trained model should consider other features when the feature $x^3$ is undefined. Thus, we can evaluate as high the plausibility of the hypothesis $H_i^3 :$ $x_i^3 \; is \; relevant$ based on the evidence $E_i \; : \; c_i^1 = 1$ while $H_i^3$ deemed implausible based on the evidence $E_i \; : \; c_i^1 = 0$.

This example illustrates how the plausibility of a hypothesis regarding the relevance of a feature, $x^j$, can be adjusted by incorporating additional external information, $c^i$. Consequently, a trained model should allocate/pay more attention to relevant features for each observation, $x_i$, in a manner that aligns with the plausibility of the hypothesis "$H \; : \; feature \; x_i^j \; is \; relevant \; for \; obs. \; x_i$" for $j \in \{1, \ldots, p\}$.

## 5 EPI-ATTENTION

### 5.1 DEFINITION

Let $X = \{x_i; i = 1, \ldots, N\} \subset \mathcal{D}$ the dataset under study and $\mathcal{C} = \{c \in \mathcal{M}; c = \chi(x)\} \subset \mathcal{M})$ the related context set.

By considering both the input sequence $x_i = [x_i^1, x_i^2, \cdots, x_i^j, \cdots, x_i^p] \in X$ and the related context $c_i = \chi(x_i)$, we propose/define the "Epi-Attention" as the set of functions $\mathcal{F}$ that map couples $(x_i, c_i) \in X \times \mathcal{C}$ to the vector $a_i = [a_i^j]_{j=1}^p \in \mathbb{R}^p$, such as $\forall f \in \mathcal{F}$ :

$$f \colon X \times \mathcal{C} \to \mathbb{R}^p$$
$$(x_i, c_i) \mapsto a_i = [a_i^j]_{j=1}^p \tag{4}$$

with : $a_i^j$ modeling the plausibility of the hypothesis $H_i^j \; : \; x_i^j \; is \; relevant \; for \; x_i$ based on the evidence $c_i$, $\forall j = 1, \ldots, p$ and $\forall i = 1, \ldots, N$.

Based on this definition, we note $a_i = f(x_i, c_i)$ the attention vector. It's composed of $p$ elements $a_i^j \in a_i$ that assess the relevance of $x_i^j \in x_i \forall j = 1, \ldots, p$ according to the provided context. Thus, it enables us to assign appropriate attention and significance to relevant features.

Consequently, a large value for $a_i^j$ enhances the relevance/importance of $x_i^j$ while a small value for $a_i^j$ diminishes/reduces its influence. As illustrated in figure 1.

Finally, an epi-vector $\tilde{x}_i = [\tilde{x}_i^1, \tilde{x}_i^2, \cdots, \tilde{x}_i^j, \cdots, \tilde{x}_i^p]$ is calculated by performing an element-wise multiplication between the attention vector $a_i$ and the input sequence $x_i$ . This operation can be expressed as:

---

[1]https://archive.ics.uci.edu/dataset/222/bank+marketing

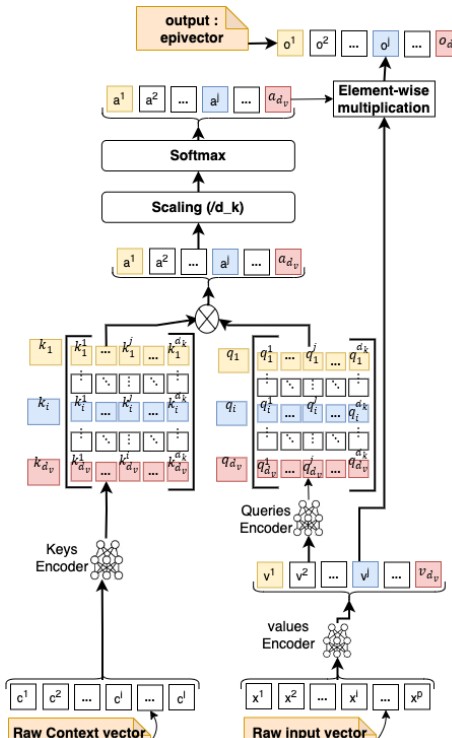

Figure 1: The Epi-Attention mechanism

$$\forall j = 1, \ldots, p, \tilde{x}_i^j = a_i^j \times x_i^j \tag{5}$$

$\forall i = 1, \ldots, N$

In summary, the objective of the Epi-Attention model is to determine the optimal attention weights for each element $x_i^j$ in the input sequence $x_i$ based on their relevance according to the context $c_i$. By strategically enhancing or silencing specific sections of the sequence $x_i$, the attention weights vector $a_i$ plays a crucial role in shaping the output, enabling dynamic and precise control over the relevance of different features.

It is worth noting when processing the raw sequence $x_i \in X$ poses challenges, particularly when dealing with mixed data types (images, words,...). The concept of epi attention remains the same by using the appropriate sequence embedding as mentionned in equation (2) and/or (3). The next section introduces our proposed implementation for effectively modeling an Epi-Attention model.

## 5.2 SCALED DOT-PRODUCT EPI-ATTENTION

To compute the attention weights $a_i = [a_i^j]_{j=1}^{d_v}$ for $i = 1, \ldots, N$, we propose a novel implementation called the "scaled dot-product Epi-Attention".

Let $x \in X$ the input sequence and $c$ the associated context. Using a neural network $net^Q$ that maps the input sequence $x$ to a matrix $Q \in \mathbb{R}^{d_k \times p}$ composed of $p$ queries vectors of dimension $d_k$ :

$$net^Q : X \to \mathbb{R}^{d_k \times p}$$
$$x \mapsto Q \tag{6}$$

with $Q = [q_1, \ldots, q_j, \ldots, q_p]^T$ and $q_j \in \mathbb{R}^{d_k}$ represents the query embedding of element $x^j \in x, \forall j = 1, ..., p$.

Similarly, the context sequence $c$ is mapped into a matrix $K \in \mathbb{R}^{d_k \times p} = [k_1, \ldots, k_j, \ldots, k_p]^T$ composed of $p$ keys through a neural network $net^K$ :

$$net^K \colon X \to \mathbb{R}^{d_k \times p}$$
$$x \mapsto K \tag{7}$$

with $K = [k_1, \ldots, k_j, \ldots, k_p]^T$ and $k_j \in \mathbb{R}^{d_k}$ represents the key embedding of the context for the element $x^j \in x, \forall j = 1, \ldots, p$.

To obtain the element $a^j$ of the epi-weights $a = (a^j)_{1 \leq j \leq p}$, we compute the dot-product of the query $q_j$ and the corresponding key $k_j$, scale it by dividing it by $d_k$ and we apply an activation function (ex. sigmoid,...etc) to normalize :

$$a^j = f\left(\frac{q_j \cdot k_j}{d_k}\right) \tag{8}$$

Finally, when processing the raw sequence $x$ is challenging, we use a neural network $net^V$ to embed the raw sequence $x$ according to formulas 2 or 3.

It is worth noting that our implementation of "the Epi-attention", depicted in figure 1 and algorithm 5, differs from the traditional Scaled Dot-Product Attention introduced in Vaswani et al. (2017). In contrast to the conventional attention mechanism, which primarily emphasizes mapping the input sequence $x$ to the output sequence $y$ using elements from $x$ as attention triggers, the Epi-Attention approach shifts its focus toward the input sequence $x$ itself. It aims to reevaluate the relevance of its elements by incorporating external information $c$. This shift in perspective allows for a more comprehensive understanding and utilization of the input sequence within the attention mechanism. Consequently, instead of generating an attention matrix, we produce a vector of dimension $d_v$ that fits the representation vector $v$. As a side benefit, this distinction allows us to transform/re-weight the input data within the same representation space $\mathcal{R}^{d_v}$ and could yield more explainable models as shown in section 6.

### 5.3 Self-Epi-Attention

In classical attention mechanisms Vaswani et al. (2017), self-attention has demonstrated its efficacy across various tasks, such as reading comprehension and learning task-independent sentence representations. In our framework, the concept of Epi-Attention is particularly valuable for assessing the consistency of the input sequence $x$. Self Epi-Attention regulates the information conveyed by the input sequence $x$ itself, enhancing the most relevant features while minimizing inconsistencies. Unlike classical self-attention, which evaluates each element in isolation, Self Epi-Attention handles conflicts by considering the impact of all elements $x - [x^j]$ (all elements but $x^j$) on the relevance of $x^j$ within the sequence $x$.

To delve into this further, we look at the hypothesis $H_i^j$ related to the $j^{th}$ element of a sequence $x_i$:

- $H_i^j : x_i^j \ is \ relevant \ for \ x_i$ (hypothesis)
- $E_i : \ c_i \ is \ internal \ information \ = \ x_i$ (evidence)

Thus, self Epi-Attention aims to preserve the internal consistency of an observation $x_i$.

## 6 Experiment and Results

This section outlines three applications where Epi-Attention and Self-Epi-Attention can be exploited to enhance model explainability and dynamically adjust feature relevance and/or facilitate processing mixed datatypes. These applications address domain-specific challenges such as :

It is worth noting that the Epi-Attention method was designed as a neural network layer to facilitate its integration into various existing models. This design choice allows researchers and practitioners to leverage the benefits of Epi-Attention within their preferred models and frameworks. By providing a dedicated layer for attention modeling, the integration process becomes straightforward, enabling researchers to explore and exploit the enhanced Epi-vector $\tilde{x}$ seamlessly offered

| App. | Domain/Dataset | task Description |
|------|----------------|-----------------|
| 1 | Medical Diagnosis (Wisconsin Breast Cancer Dataset) | Apply Self-Epi-Attention to capture internal feature correlations for improved explainability in medical diagnoses. Self-Epi-Attention adjusts feature relevance based on internal relationships like nuclear texture, allowing the model to align feature importance with medical knowledge. |
| 2 | Marketing (Bank Marketing Dataset) | Use Epi-Attention to adjust numerical feature importance based on categorical features to predict customer behavior. Epi-Attention dynamically weights features like "balance" and "contact duration" by factoring in external context such as previous contacts, improving the model's logic and explainability. |
| 3 | Neuroimaging (Autism Spectrum Disorder, Multi-Site Data) | Leverage Epi-Attention to handle site-specific variability in multi-site neuroimaging data. Epi-Attention adjusts feature relevance based on the site of origin, accounting for variations in scanning protocols and demographics, offering a more accurate and understandable model. |

Table 1: Applications of Epi-Attention and Self-Epi-Attention across different domains

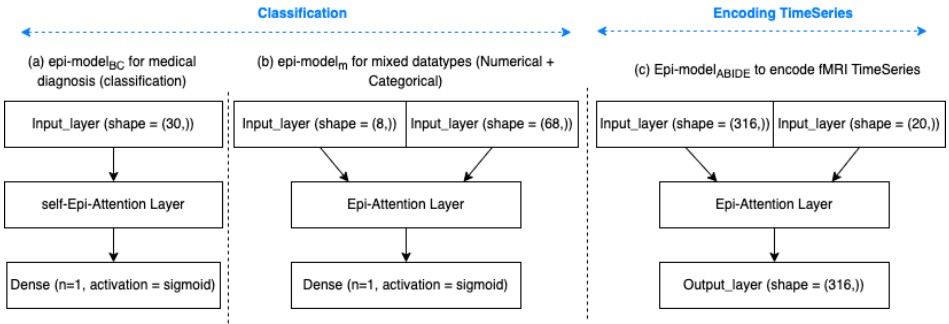

Figure 2: Architectures of the three used epi-models, from the left to the right : $epi - model_{BC}$ for medical diagnosis with Wisconsin Breast Cancer Dataset, $epi - model_m$ for mixed datatypes handling with Bank-marketing Dataset, $epi - model_{ABIDE}$ for multi-site timeseries Analysis with ABIDE-II

by Epi-Attention. The code to reproduce all results of this section is at the GitHub repository https://github.com/XXX.XXX.

## 6.1 APPLICATION 1: MEDICAL DIAGNOSIS (WISCONSIN BREAST CANCER DATASET)

We propose $epi - model_{BC}$ represented in figure 2 for medical diagnosis with the Wisconsin Breast Cancer dataset, a neural network designed to dynamically adjust feature relevance using Self-Epi-Attention. The dataset contains 569 samples, each described by 30 numerical features extracted from images of breast tissue. The task is to classify each sample as either benign or malignant.

### 6.1.1 MODEL ARCHITECTURE ($epi - model_{BC}$):

- Input Layer: The first layer that accepts the 30 features from the dataset.
- Self-Epi-Attention Layer: The second layer, which applies Self-Epi-Attention to dynamically adjust the relevance of each feature based on internal correlations. This layer enables the model to emphasize or silence certain features depending on the class-specific properties of the data.
- Output Layer: A single neuron with a sigmoid activation function, which outputs the probability of a tumor being benign or malignant.

This architecture is specifically tailored to leverage the power of Self-Epi-Attention in medical diagnosis, allowing the model to focus on the most relevant features for each class, improving model explainability and keeping the same classification performance.

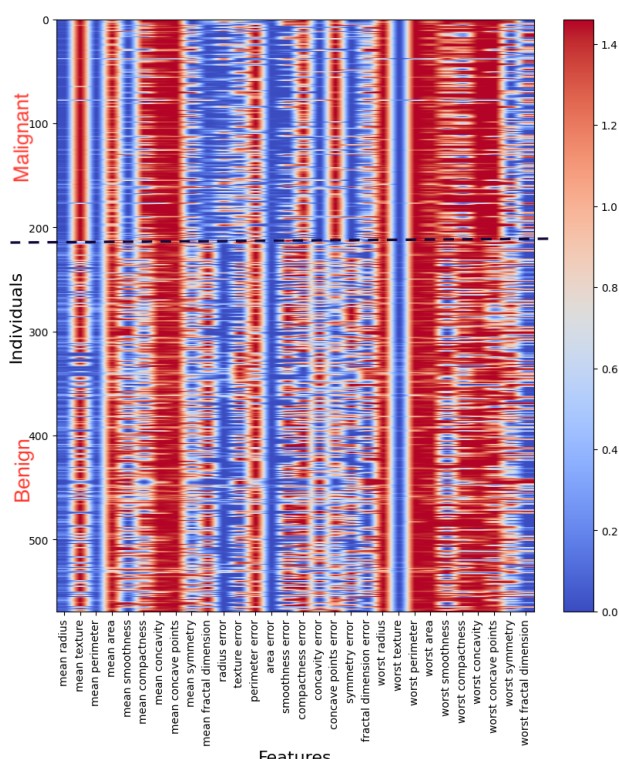

Figure 3: features importance for the Breast Cancer Dataset 569 patients (212 malignant (M) tumors vs 357 benign(B)) obtained from self-epi-attention layer 1 of $epi - model_{CB}$

### 6.1.2 ANALYSIS OF FEATURE RELEVANCE IN THE WISCONSIN BREAST CANCER DATASET USING SELF-EPI-ATTENTION

Self-Epi-Attention generates feature relevance vectors $a_i = [a_i^j]_{j=1}^{30} \in \mathbb{R}^{30}$ for each data sample, as illustrated in the accompanying figure 3. The dataset is ordered with malignant cells in the first 212 rows and benign cells from row 213 to the end (357 rows). Experts in oncology often focus on class-specific properties Street et al. (1993); Wohl et al. (2023); Chitalia & Kontos (2019), and Self-Epi-Attention enables the model to emphasize local correlations, dynamically modulating feature importance for each class. This allows the model to capture relationships that global models typically miss. Key observations include:

- The role of Texture in Malignant Diagnosis: The feature mean_texture receives significantly higher attention for malignant cells, highlighting its importance in describing malignant tumors. In contrast, it receives less attention in benign cell diagnosis, which aligns with domain knowledge about the role of texture in cancer diagnosis Chitalia & Kontos (2019).

- Surface Features in Benign Cells: On the other hand, features related to surface properties, such as mean_fractal_dimension, mean_smoothness, and mean_symmetry, are more relevant and get higher attention in benign cells, while playing a lesser role in malignant diagnoses Wohl et al. (2023).

These observations confirm that Epi-Attention successfully captures the feature importance variability across different classes, dynamically reassessing feature relevance based on the given case.

Beyond focusing on individual classes, Self-Epi-Attention allows the model to adjust feature importance across subpopulations within the data, it ensures that feature weighting does not disproportionately favor the most represented class or subpopulation—an issue often seen with global models Ünalan et al. (2024).

## 6.2 APPLICATION 2 : MIXED DATA-TYPES IN TELEMARKETING WITH BANK MARKETING DATASET

The Bank Marketing Dataset is widely used to predict whether clients will subscribe to a term deposit. The dataset contains both categorical features (e.g., job type, poutcome) and numerical features (e.g., euribor3m, age, duration of the last call), making it an ideal test case for handling mixed data-types.

This dataset is also highly imbalanced, with only around 11% of clients subscribing (success) and 89% (36000 rows) not subscribing (fail). Handling this imbalance effectively is critical to ensuring the model captures relevant patterns in the minority class (subscribers) and for this task we developed the epi-model ($epi - model_m$ see figure 2) to perform the classification task

### 6.2.1 MODEL ARCHITECTURE ($epi - model_m$):

- Input Layers: Two input layers handle the mixed data-types: One input layer processes one-hot encoded categorical features. Another input layer processes numerical features.
- Epi-Attention Layer: This layer dynamically adjusts the relevance of numerical features, allowing the model to emphasize features based on the context within the categorical features dataset.
- Output Layer: The final layer predicts whether the client will subscribe to the term deposit based on the dynamically reweighted features from the Epi-Attention layer.

Analysis of Feature Relevance Using Epi-Attention Figure 4 illustrates the feature importance for numerical attributes in the Bank Marketing Dataset, derived from the Epi-Attention layer of $epi - model_m$ (with 91% of accuracy). By using Epi-Attention, the model is able to adapt feature importance dynamically for both the majority and minority classes, handling the imbalance in the dataset more effectively than traditional models. This leads to improved explainability and ensures that the model captures critical patterns across all clients, particularly in the underrepresented class with several heterogeneous patterns.

## 6.3 APPLICATION 3 : CONTEXTUAL EMBEDDING IN NEUROIMAGING WITH ABIDE-II DATASET

The Autism Brain Imaging Data Exchange II (ABIDE-II) dataset is a large-scale neuroimaging dataset designed to study autism spectrum disorder (ASD) across different populations and sites. The dataset contains functional and structural MRI data collected from 20 international sites, providing a unique opportunity to explore variability in neuroimaging data due to differences in scanning protocols and demographic distributions. For this application, we implemented ($epi - model_{ABIDE}$) in figure 2 to encode timeseries from (fMRI scans) while incorporating the context of the originating site as an additional input. This model was trained using a Siamese architecture to capture meaningful site-specific embeddings for time series comparison.

### 6.3.1 MODEL ARCHITECTURE ($epi - model_{ABIDE}$):

- Input Layers : One layer (1) processes a time series of 316 timestamps, representing fMRI data while the another Layer (2) accepts a one-hot encoded site provenance vector of shape (20,), corresponding to the 20 different data collection sites.
- Epi-Attention Layer: The second layer, which applies Epi-Attention to dynamically reweight the time series features (timestamps) based on the site context. This allows the model to adjust its feature relevance following the specific differences of each site.

The output time series embeddings are then compared in a siamese framework, with the goal of learning robust representations that factor in site-specific variability.

### 6.3.2 FOCUS ON SITE EMBEDDINGS:

While previous applications focused on the attention paid to specific elements, this application emphasizes the embeddings generated for the 20 source sites. Figure 5 shows the first principle plan

of A PCA applied to the learned embeddings revealed clusters that naturally reflect similarities between the protocols followed by different sites. The clustering shows how Epi-Attention captures the intrinsic relationships between sites, based on neuroimaging protocols and demographic variations. These results align with known site characteristics Nielsen et al. (2013); Kunda et al. (2020).

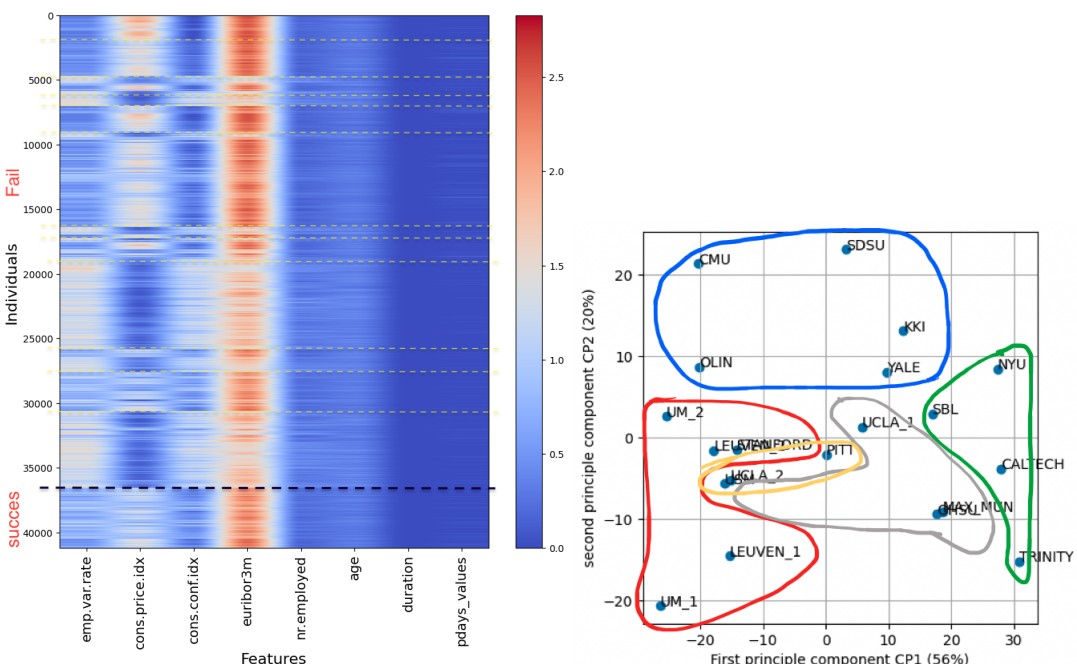

Figure 4: Features importance for numerical attributes of the unbalanced Bank Marketing Dataset (41,188 clients, 11% positive answers) obtained from epi-attention layer 1 of $epi-model_m$.

Figure 5: The first principal plan from PCA on sites encoding with net-K from the $epi-model_{ABIDE}$

## 7 CONCLUSION AND FUTURE WORK

In this work, we proposed Epi-Attention, a novel attention mechanism that dynamically adjusts feature relevance by leveraging both internal feature correlations and external contextual information. The key innovation of Epi-Attention lies in its ability to tailor feature weighting not just globally across a dataset but dynamically at the instance or group level, where contextual relevance becomes paramount. Unlike traditional models, which often apply static importance weights, Epi-Attention enables a more nuanced, context-aware feature relevance evaluation that adapts to local correlations and external signals.

Also, we showed how this concept of context-driven feature relevance can be applied to a variety of challenging real-world scenarios. The core contribution of Epi-Attention lies in its ability to offer dynamic feature relevance evaluation, where the importance of features is not fixed but evolves according to the specific characteristics of the data and the context in which it is situated. This makes Epi-Attention especially powerful in scenarios where feature importance varies across subpopulations, conditions, or instances.

Looking forward, future work could expand on the use of Epi-Attention in other domains and data modalities, further exploring its potential for enhancing model transparency and context-driven decision-making. The promising results shown in this paper point to a broader application of context-aware attention mechanisms in machine learning, especially in fields where explainability is crucial.

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
