# OpenReview forum: "Epi-attention : Adaptive Context-Aware Attention for Dynamic Feature Relevance in Neural Networks"
_ICLR.cc/2025/Conference — ICLR 2025 Conference Withdrawn Submission_

### Official Review · Reviewer_TfJE · 2024-10-20

**Soundness:** 1
**Presentation:** 2
**Contribution:** 1
**Rating:** 3
**Confidence:** 3

**Summary:**

This paper proposes Epi-Attention. A "context-aware" attention mechanism designed to enhance the relevance/importnace of featuresby incorporating "external contextual" information. It applies the introduced attention mechanisms to three datasets.

**Strengths:**

- Definitions etc. are consistent and correct

**Weaknesses:**

- It seems like an unfinished paper with many small mistakes. For example, the model figure is obscured, the GitHub repository link is faulty, and the writing lacks polish, with several grammar mistakes. E.g. citations in the related work section are wrong.
- Since the GitHub link is incorrect, no code is provided.
- The experiments are entirely missing, leaving no way to assess whether epi-attention has an effect on performance. Since the authors showcase 3 datasets where I guess (binary) cross entropy is minimized, AUC scores on a held-out-test set should agt least be reported.
- The figures aim to showcase the advantages of Epi-attention, but they appear more anecdotal, as the model is tested on only a few datasets and no ground truth is provided. The results presented seem plausible to the authors but are not fully substantiated. Either a user-study or a simple ablation study could strengthen the arguments.
- There is no comparison to standard attention scores. Given that the authors have chosen tabular problems, a comparison with FT-Transformers and the importance scores via the CLS token should be included.

**Questions:**

- Why not just using the "context" as part of the sequence? Given your example on page 4 (lines 173-174), why not just use c^1 as an additional feature? I would assume if you do so and use a simple architecture like the FT-Transformer, the performance results would be better than with the proposed method.

---

### Official Review · Reviewer_Ajbg · 2024-10-27

**Soundness:** 1
**Presentation:** 1
**Contribution:** 1
**Rating:** 3
**Confidence:** 4

**Summary:**

This paper investigates the problem of context-aware attention and introduces Epi-Attention, which aims to dynamically adjust feature relevance by incorporating external contextual information into neural networks.

**Strengths:**

Context-aware attention is an interesting and important research problem with broad applicability across various domains.

**Weaknesses:**

1) The literature review is insufficient, as it omits key works [1, 2, 3] that are fundamental to the development of context-aware attention mechanisms. The authors should explicitly discuss how Epi-Attention and Self-Epi-Attention differentiate themselves from these methods, particularly in their approach to integrating contextual information. A comparative analysis highlighting these differences is essential to better position Epi-Attention within the existing body of work.
[1] "Context-Aware Self-Attention Networks", AAAI, 2019
[2] "A Context-Aware Attention Network for Interactive Question Answering", SIGKDD, 2017
[3] "Context-Aware and Time-Aware Attention-Based Model for Disease Risk Prediction With Interpretability", TKDE, 2023

2) The Epi-Attention mechanism in this paper lacks clear innovation compared to existing context-aware attention models such as context-aware self-attention and multi-modal mechanisms. Further clarification on how Epi-Attention integrates external or internal contexts or how it surpasses these existing models would strengthen the paper.

3) The paper lacks enough experiments, and the datasets used are quite limited. Additionally, the results lack detailed quantitative analysis. The authors should include more comprehensive performance metrics and provide comparisons with baseline models, such as [1, 2, 3], and Tri-Attention, across related domains like NLP tasks, including retrieval-based dialogue, sentence semantic matching, machine reading comprehension, and machine translation.

4) There are too many writing issues throughout the paper, including spelling errors, grammar mistakes, and unclear explanations. e.g., (1) "efficency" in line 029; (2) In the phrase "in Addition, explainability is crucial...," the word "Addition" should not be capitalized (Line 048); (3) "mentionned" should be corrected to "mentioned" (Line 255). (4) Many figures (e.g., Figure 1) are difficult to read due to poor formatting and labelling. (5) The code link (in line 358) is not yet prepared.

**Questions:**

Please address the above weaknesses.

---

### Official Review · Reviewer_niFW · 2024-10-28

**Soundness:** 2
**Presentation:** 1
**Contribution:** 1
**Rating:** 1
**Confidence:** 4

**Summary:**

This paper propose EPI-ATTENTION, which aims to learn attention weights for given input x and context c. The clarity and novelty of the paper should be significantly enhanced.

**Strengths:**

1. The paper propose to model attentions by incorporating contexts

2. The paper targets real-world datasets and applications

**Weaknesses:**

1. The contribution of the paper is not clear. The paper looks very similar to vanilla transformer, the only difference, as the author stated in line 288, is the incorporation of contexts, or meta-data into representation learning. How is this very much different from concatenate contexts to the input and get the representation?

2. The claimed extra interpretability and efficiency are already well-known for models like transformers, and thus are not very interesting.

3. The referenced paper are relatively old and are not enough for comparison for novelty.

4. The writing and presentation of the paper are of poor quality. The writings are not clear. The equations and images leave huge margins. The resolution of the images are low. Some figures, like figure 5, is hard to read.

**Questions:**

Considering the limited technical scope, novelty and contribution, I recommend the authors to evaluate their work for publication, especially at top conferences like ICLR.

---

### Official Review · Reviewer_8a1s · 2024-11-01

**Soundness:** 2
**Presentation:** 1
**Contribution:** 2
**Rating:** 3
**Confidence:** 5

**Summary:**

In this paper, the authors propose a context-aware attention mechanism called Epi-Attention to enhance the relevance of features in neural networks by incorporating external contextual information. By leveraging the dynamic aspect of Epi-Attention, models can highlight local correlations that are characteristic of certain classes, offering a more transparent and interpretable decision-making process compared to global correlations favorized by classical approaches such as Decision trees, Logistic regression and Neural Networks. Experiments on three public datasets from different areas demonstrates the effectiveness for enhancing explainability.

**Strengths:**

(1)	A new attention architecture is proposed.
(2)	Experiments are done on three real-world datasets from different areas.

**Weaknesses:**

(1)	The paper is poorly written, and hard to follow.
(2)	The figure quality is low, especially Figure 1.
(3)	The challenges 1 and 2 are not solid, for there are many local interpretation methods, such as the instance-wise interpretation of neural networks, and some methods based on attention.
(4)	There are more methods based on attention, but they are not discussed.
(5)	How to collect the context data is not clear.
(6)	I don’t see how the evaluate the effectiveness of the proposed method in experiment.
(7)	No performance comparison against existing interpretation methods is done.

**Questions:**

(1)	How to collect the context data?
(2)	How the evaluate the effectiveness of the proposed method?

---

### Note · Authors · 2024-11-14

I have read and agree with the venue's withdrawal policy on behalf of myself and my co-authors.